# Lungfish and the Long Defeat

**Anne Kemp**

Natural History Museum, Rue de la Croix-Rouge 4, 3983 Geneva, Switzerland;
annerkemp@gmail.com or a.kemp@uq.edu.au

**Abstract:** Australia has an excellent fossil record of lungfish that begins in the Devonian and includes many species in Tertiary and Quaternary deposits. The extant Australian lungfish, *Neoceratodus forsteri*, occurs in Pliocene deposits, but is now restricted to a handful of coastal rivers in Queensland. Some of the fossil taxa, belonging to species related to *N. forsteri*, are represented by only a few specimens, but others include large numbers of tooth plates. The existence of these taxa, even if they are represented by only a few specimens, indicates that lungfish were present in lakes and rivers in central and northern Australia in the past, and that the potential habitats for these fish were more extensive then than they are now. Many of the fossil populations died out because Australia became more arid, and the remaining species became isolated in large river systems in the north and east of the continent. However, the cause of extinction of some fossil populations was not always related to increasing aridity. Several fossil populations were apparently living in poor conditions. They stopped spawning and adding new members to the population. The remaining individuals showed advanced age and many diseases before the population disappeared. This can be observed in the present day, and one population in an isolated reservoir is already extinct.

**Keywords:** fossil lungfish; living lungfish; habitats; extinction

## 1. Introduction

Lungfish have an almost uninterrupted history in Australian fossil sites. Species of note occur in every geological period except the Jurassic, when conditions were unsuitable for the preservation of vertebrate bone. The Tertiary and Quaternary deposits of Australia contain many species [1,2]. The decline of the fossil populations has implications for the future of the last remaining lungfish in Australia, *Neoceratodus forsteri* [3], the only surviving representative of the Neoceratodontidae. The history of these species, their morphology and their eventual decline, has implications for the current situation of the living Australian lungfish. The evolutionary lineage related to *N. forsteri* survived and radiated into new environments in the Tertiary and flourished for a while in Australia, before declining to a single species that is confined to coastal rivers in southeast Queensland. There is only one reliable fossil record of lungfish in coastal deposits, but *N. forsteri* still lives in coastal rivers and lakes now. *Neoceratodus* is not recorded from other continents, although related groups taxa, such as *Metaceratodus*, were located in Australia, as well as in Africa and South America, where other species of lungfish, *Protopterus* and *Lepidosiren*, have survived. Species of *Metaceratodus* are now extinct in these continents and in Australia [2].

Lungfish fossils have been described from deposits of rivers, lakes, and wetlands in central, northern, and eastern Australia. Although many of the deposits contain only a few tooth plates, of one or two species, they indicate that habitats suitable for lungfish were present in many parts of central and eastern Australia, and lungfish were actually widespread and not isolated, as they are now, in a handful of coastal habitats in southeast Queensland. Some of the fossil populations were healthy, with a range of different species, with young and adult lungfish in the population. Others were actively spawning, and the dentition has few pathologies, but the population rarely included large specimens, suggesting some limiting factor in the habitats where the lungfish lived. Still others had

only large tooth plates afflicted with a number of diseases that can be related to age and a poor diet, and there was no indication that young fish were present. More potentially damaging for the living lungfish populations, characteristics of the dentitions of the fossil populations that became extinct because of disease and environmentally related damage, as well as a failure of recruitment, are mirrored in the conditions found in remnant populations of the living lungfish, *N. forsteri*. Rivers and reservoirs where the lungfish now live have been damaged by drought, floods, and the building of reservoirs over the rivers. The loss of biodiversity has removed food supplies for adult and young lungfish, as well as damaged refuges for vulnerable hatchlings [4]. Spawning has ceased in many places, and recruitment of young fish to the adult population has been affected [5,6].

Analysis of tooth plate characters can help to determine the way the fish has lived, how well it has fed, whether it ate soft or hard food, and if there was active recruitment. It can also reveal the lack of food. Collation of the data on individual teeth allows us to assess the characters and the quality of the environment in which a population of lungfish lived [5,7]. Because the tooth plates are not repaired or replaced, and grow slowly from the enamel/bone junction and the pulp cavity, disease conditions that affect the fish leave traces in the tooth plates, as does the effect of age. This means that tooth plates can be used to assess details of the population structure of a group of lungfish and the health of individual animals. Use of this data explains that some populations of lungfish became extinct as a result of lack of food, failure to recruit young fish, and increasing age and disease. Similar problems affect lungfish in habitats in southeast Queensland at the present time [4].

## 2. Materials and Methods

### 2.1. Structure and Growth of the Dentition

Tooth plates were assessed for biological age, normal wear, and disease characters (Figures 1–3). Specimens of the living lungfish are either part of the collections of the Queensland Museum (annotated as QMI) or part of my personal collection (annotated as AN). Fossil specimens come from the collections of the Queensland Museum (annotated as QMF).

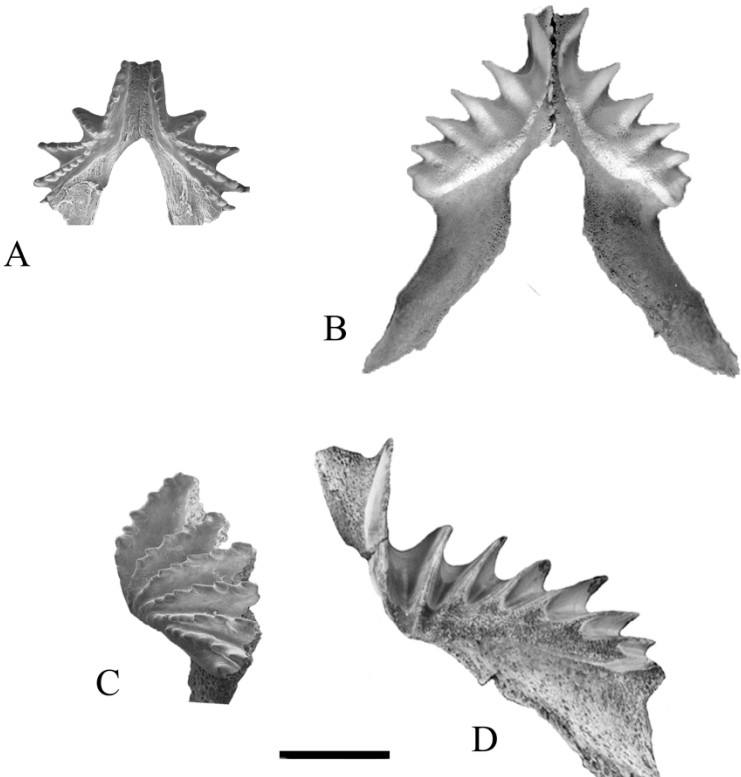

**Figure 1.** *Cont.*

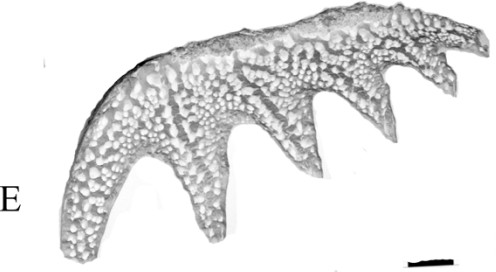

E

**Figure 1.** Changes in the tooth plates as lungfish develop. (**A**) Prearticular tooth plates from a juvenile *Neoceratodus forsteri*, AN 58-1999, raised in the laboratory from an egg collected in the Brisbane River in 1992. (**B**) Prearticular tooth plates from an adult *N. forsteri*, QM 26014, collected in the Brisbane River in 1974. (**C**) Upper tooth plate from a juvenile *Mioceratodus diaphorus*, QMF 44425, from the Carl Creek Limestones in Riversleigh, Miocene. (**D**) Upper tooth plate of a specimen of *M. diaphorus*, QM 11023, from Frome Downs, South Australia, Oligocene. Changes in the permanent dentition are dramatic in specimens of the family Neoceratodontidae as the fish develop, but this does not mean that they come from separate species. (**E**) A scanning electron micrograph of the pulp cavity surface of a prearticular tooth plate of *Mioceratodus anemosyrus*, QMF 51703, from Riversleigh, showing cores of petrodentine running along each ridge, flanked by interdenteonal dentine. Scale bars = 1 cm.

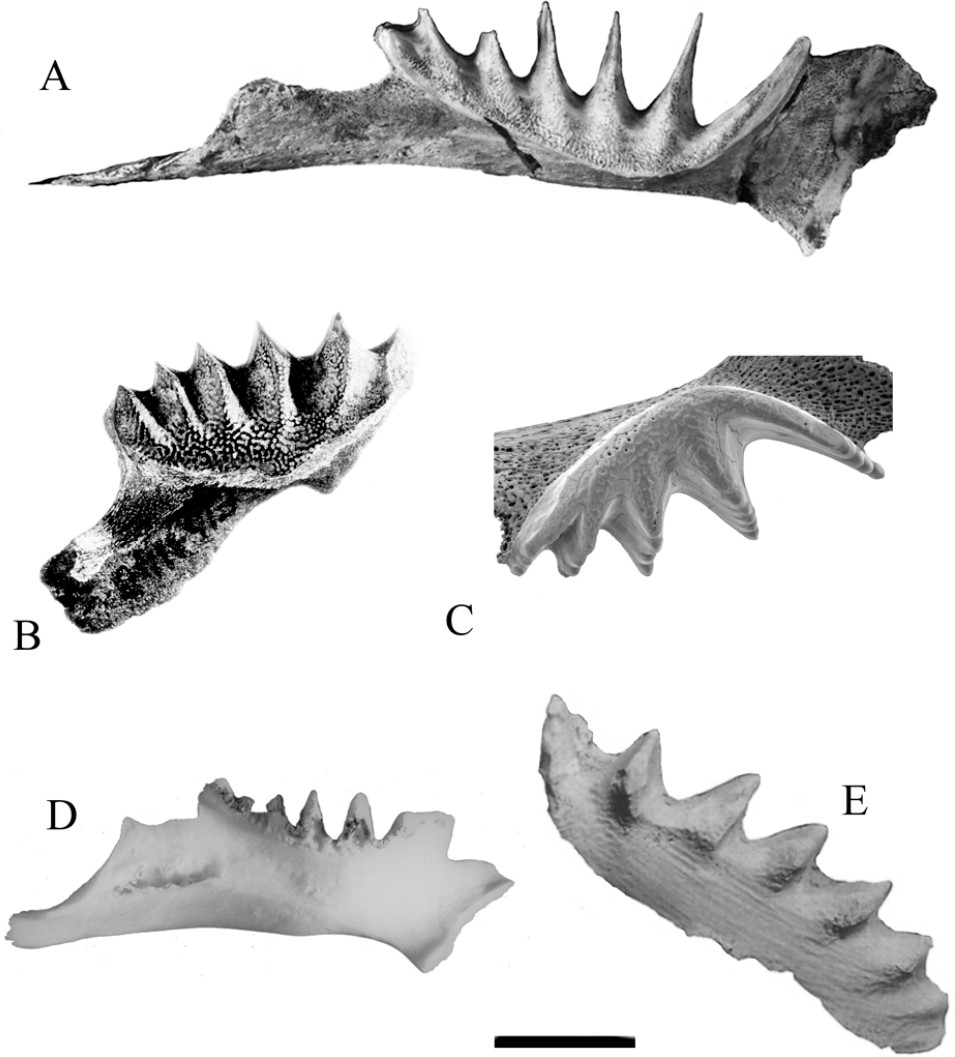

A

B

C

D

E

**Figure 2.** Normal wear in lungfish tooth plates. (**A**) Occlusal view of a large tooth plate of *Mioceratodus poastrus* AMNH 11322 with shallow interridge facets. (**B**) "*Ceratodus forsteri*" de Vis QMF 15009, post

Pliocene, a tooth plate with deep grinding wear. (**C**) A juvenile lower tooth plate of *N. forsteri*, AN 25998, from the Brisbane River with normal wear. (**D**) Medial view of a tooth plate from Enoggera reservoir with deep facets resulting from the crushing of hard items of food, AN88-109. (**E**) Severe attrition in a tooth plate of *Archaeoceratodus theganus*, MV P 160524. Scale bar = 2 cm.

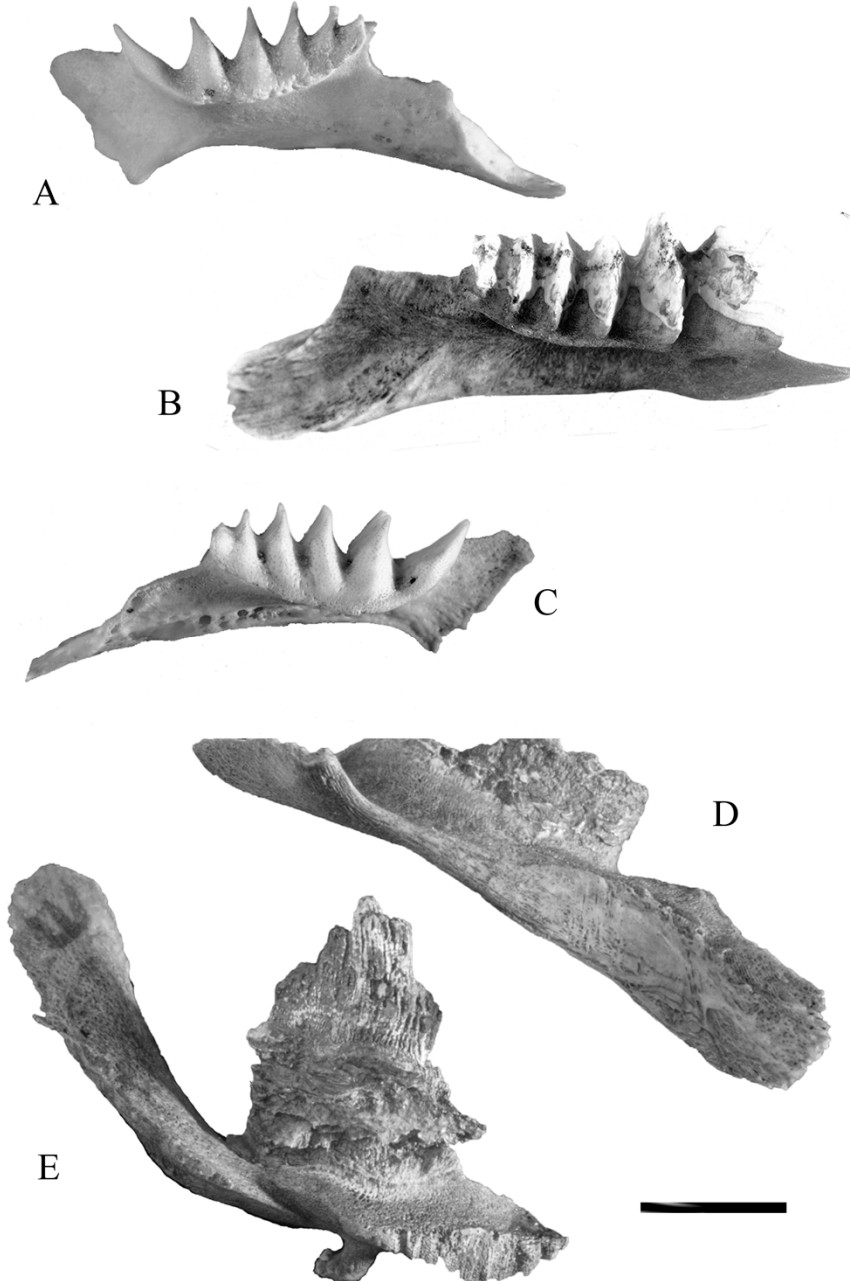

**Figure 3.** Pathological conditions in lungfish tooth plates and bones, all illustrated in specimens of *Neoceratodus forsteri*. (**A**) A tooth plate from Enogerra Reservoir showing erosion of the mediolingual face of the tooth plate AN 88-110. (**B**) Labial face of a tooth plate from Enoggera Reservoir with carious lesions. AN88-14. (**C**) Jaw bone and tooth plate from Enoggera Reservoir with osteoporosis AN 09-112. (**D**,**E**) Matching upper and lower tooth plates of a fish from Enoggera Reservoir with serious hyperplasia. QMI 26010. Scale bar = 2 cm.

## 2.2. The History of Fossil and Living Lungfish Populations

The current distribution of the living lungfish is shown in Figure 4. Tertiary and Quaternary localities where fossil and living lungfish are found are shown in Figure 5.

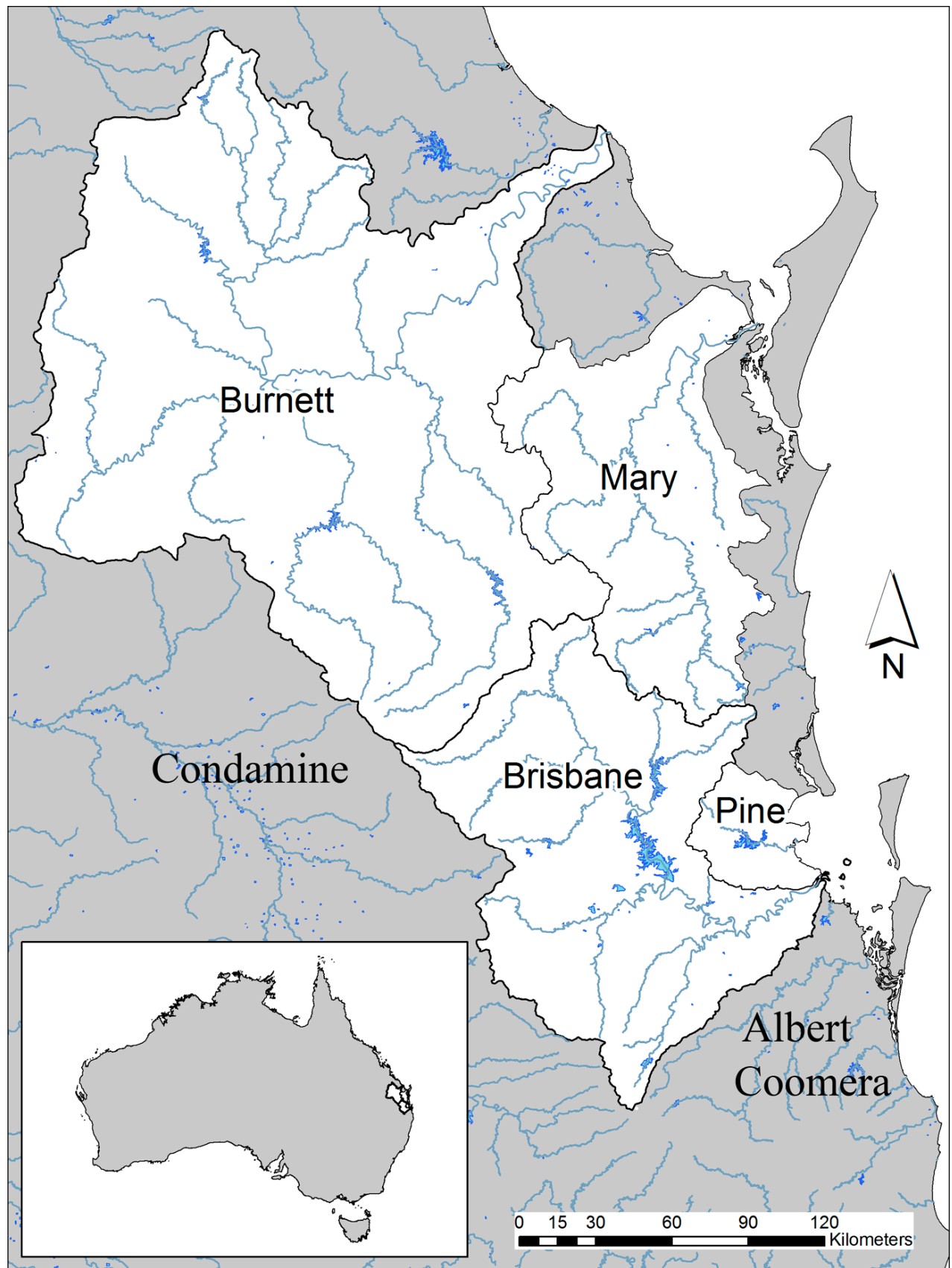

**Figure 4.** Map showing endemic and translocated river sites where lungfish are, or have been, present.

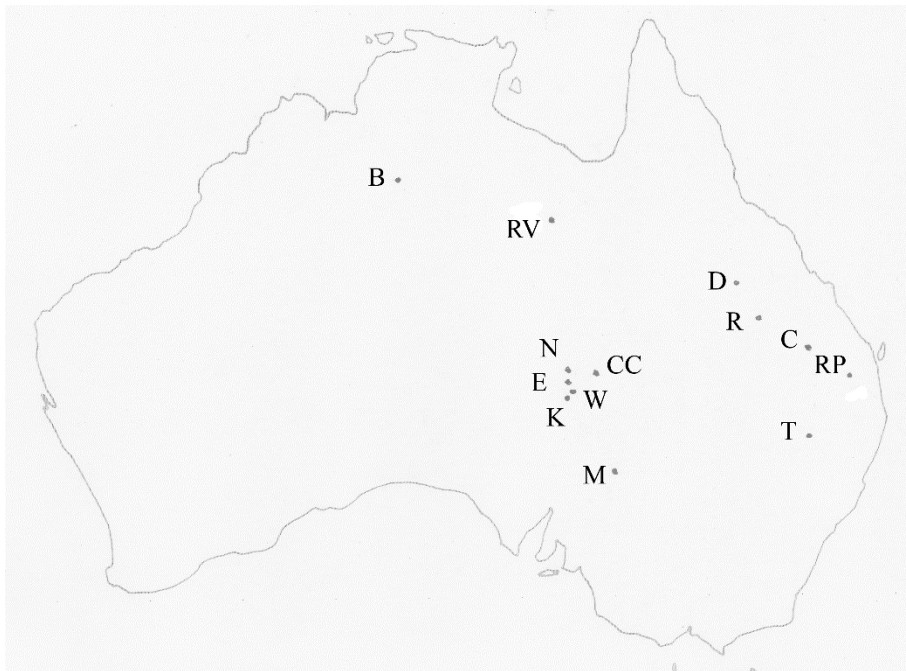

**Figure 5.** Tertiary and Quaternary fossil deposits in Australia. B. Bullock Creek. C. Chinchilla and King Creek. CC. Cooper Creek. D. Duaringa. E. Etadunna. K. Katapiri. M. Moorna. N. Namba. R. Rundle. RV. Riversleigh. RP. Redbank Plains. T. Toorale. W. Wipijiri.

## 3. Results

### 3.1. Structure and Growth of the Lungfish Dentition

In most lungfish, including all of the Neoceratodontidae, the dentition consists of permanent tooth plates with persistent cusps, separated early in development (Figure 1A,C), which grow and form ridges that are arranged in a radiating pattern (Figure 1B,D).

Fusion of the cusps to each other and to the underlying jaw bone produces a tooth plate, based on a template of mantle dentine that surrounds bone trabeculae that are enclosed within the tooth plate [8]. The mantle dentine is covered by enamel. The framework of enamel, mantle dentine, and bone surrounds two further forms of dentine, known as interdenteonal and circumdenteonal dentines, which grow from within the pulp cavity below the tooth plate, and the whole structure is supported by the underlying bone of the jaws. The tooth plates expand in area and in depth without loss of structural integrity or evidence of macroscopic resorption of dentines or of enamel. Increases in the size and changes in the shape of lungfish tooth plates is actually achieved by a process involving microscopic remodelling of the bone contained within the margin of each tooth plate, and the later addition of new mantle dentine and enamel around the bone, and of circumdenteonal dentine and interdenteonal dentine within the pulp cavity. The growth process proceeds in line with the growth and remodelling of the jaw bones attached to the tooth plates [8], and the tooth plates wear continuously from the occlusal surface. Development similar to that of the living lungfish, *N. forsteri* (Figure 1A,B), can be seen in the fossil members of the Neoceratodontidae, such as *Mioceratodus diaphorus* (Figure 1C,D). Species of *Mioceratodus* and *Archaeoceratodus* have an additional form of dentine, known as petrodentine (Figure 1E), originally described in the dentition of the South American and African lungfish [9]. Petrodentine also grows from the pulp cavity [10]. Apart from the occlusal surface, the tooth plates are surrounded by epithelium, and material can become trapped between the epithelium and the tooth plate. This may cause damage to the dental tissue [7].

Lungfish cannot repair injury to the dentition [11]. The damaged tissue can possibly grow out and wear away, but if it is severe, the broken tooth plate may become infected. Neither can the fish repair the effects of age in the teeth or in their bones, such as osteopenia

or osteoporosis. Heavy, harsh wear on the tooth plates, osteoporotic bones, and large size are characteristics of an old fish, and strong bones with small tooth plates having numerous cusps on the labial face of the ridges and smooth wear suggest a young lungfish. Because the tooth plates are permanent and grow continuously, they do provide a history of the fish—not a precise, statistically verifiable history, but an indication of how the tooth plates were used and for how long. Analysis of tooth plate characters can show the way the fish has lived, how well it has fed, and whether it ate soft or hard food. It can also indicate the lack of food. Because the tooth plates are not repaired or replaced, and grow slowly from the enamel/bone junction and the pulp cavity, disease conditions that affect the fish leave traces in the tooth plates, as do the effects of age.

Grinding and crushing abrasion, and the amount of wear on the occlusal surface, are all a result of normal use of the tooth plates (Figure 2A–C). Abrasion happens when food is processed between the tooth plates. It can reveal what sort of food the fish have been eating, rough or soft, and whether they simply crushed it, or if they chewed it with a grinding movement for a while. Crushing and grinding leave different traces on the tooth plates (Figure 2), as does the length of time the fish used the teeth. In other words, wear on the tooth plates, while not a precise indication of age in years, can suggest whether a fish was young, middle aged, or old, and how it used the dentition. Grinding abrasion is assessed by the rounded shape of the inter-ridge furrows (Figure 2A–C), as well as the lack of complete incision of the furrows on the mediolingual face of the tooth plate. The ridge crests may be rounded or faceted if grinding abrasion has occurred. When pits or sharp facets are found in the inter-ridge furrows, crushing abrasion was the predominant mode of tooth use (Figure 2D). If the fish has used crushing movements for a long time, as in living lungfish from Enoggera Reservoir, the furrows are deep and extend to the mediolingual face of the tooth plate. The degree of wear, heavy or light, can be estimated by the depth of the inter-ridge furrows.

Spur and step wear, which results in the formation of a high spur on the posterior margin of the lower tooth plate and a step on the corresponding upper tooth plate, is found in large tooth plates and indicates that the diet may have included rough food. Spur and step wear is not pathological, but it does suggest prolonged trituration of harsh food and can be an indicator of age.

Attrition (Figure 2E) occurs in tooth plates as well, and this suggests that the lungfish was not feeding, but simply grinding the tooth plates together without any food. Attrition leaves the tooth plate flat, with no marked ridge crests on the occlusal surface, and sometimes with lines worn along the tooth plate. This happens in living and fossil material, and is unlikely to have had anything to do with taphonomy. It is very rare among fossil material.

### 3.2. Pathologies Present in the Dentition

Some changes to the dentition are the result of pathological processes (Figure 3). Erosion of the medial face of the tooth plates is a consequence of age and a poor environment (Figure 3A) and suggests that the fish lived in acidic water. Caries, sometimes progressing to an abscess, develops when food items are trapped under the epithelial trough, allowing the food to be held against the tooth and setting up decay processes (Figure 3B). Osteopenia and osteoporosis appear as the fish gets older (Figure 3C–E). Malocclusion can result from traumatic lesions to the tooth plate. Hyperplasia is a consequence of breakage of part of one tooth plate, permitting unfettered growth of the normal tooth plate (Figure 3D,E). It can reach extreme proportions. Jaw bones and tooth plates may be invaded by a nematode worm, resulting in permanent damage to the dentition [7]. Diseased specimens are rare in some fossil deposits, but are very common in others, and are also prevalent in some habitats of the living species.

### 3.3. Endemic Populations of the Living Lungfish

There are three, and possibly four, endemic environments for the living lungfish, *N. forsteri* (Figure 4). The Burnett River is the "type locality" where the lungfish first came to be known to scientists [3], and the Mary River is the source of several more specimens that were sent to London for further analysis [12]. The Brisbane River, where a number of lungfish bones, dating back to 3500 years BP, were collected from a rock shelter occupied by Indigenous people [13], has always had a population of lungfish. The species may also have been present in the Pine River system to the south of the Brisbane River prior to the translocation efforts of O'Connor [14]. The living Australian lungfish is now confined to coastal rivers on the eastern side of the Great Dividing Range, and is no longer found in the Murray/Darling catchment, where fossil specimens of *N. forsteri* have been collected.

### 3.4. Translocated Populations

Translocated populations of lungfish from the Mary River occur in the Coomera, Albert, and Condamine River systems [14–16], and in Lake Manchester in the Brisbane River valley, translocated from the Burnett River (Figure 4). Lungfish placed in Enoggera Reservoir, also from the Mary River, have recently become extinct, and they have also died out in Gold Creek Reservoir nearby. It is difficult to believe the story that lungfish were only found in the Mary and Burnett Rivers at the time of their discovery by scientific investigators in 1870, although this is still an article of faith with many in the Queensland and the Federal Governments [17] and frequently included in scientific papers on the species [18,19]. It is possible that lungfish were always to be found as natural populations in the Brisbane River system, and perhaps the Pine River, as well as the Mary and Burnett Rivers, and there is certainly no doubt that the distribution in historical times was restricted, as it is now. In view of the wider number of habitats where *N. forsteri* was to be found in the Pliocene and Pleistocene in eastern Australia, the distribution of the living species in historical times, whether two or four river systems were involved, was a cause for justifiable alarm, even when they were first described by scientists. Unfortunately, translocations were performed before the complete distribution of living lungfish in the southeast corner of the state was fully understood.

In 1885, Sir Ferdinand von Muller wrote to the recently formed Royal Society of Queensland, suggesting that the Society should try to "preserve our interesting dipnoi from extinction". Thanks to the persistent efforts of a member of the Society, Dr. Joseph Bancroft, the council of the Royal Society appealed to the Government for assistance, and funding was provided to collect and transfer specimens to new environments. Miva, on the Mary River, was chosen as the source of lungfish, and Daniel O'Connor, a retired banker with extensive, if rather unsuccessful, experience in translocating salmon, trout, and gourami to Queensland habitats, was asked to undertake the work.

O'Connor collected 109 fish, all but 5 from Miva on the Mary River. A few came from Munna nearby [14]. He comments that the lungfish were "fairly abundant", although the significance of this for the presumed imminent extinction of the species seems to have been missed by O'Connor and by those that supported the work in the first place. According to his own published letters and papers [14], O'Connor's specimens were all large adult fish, 39–45 inches in length, and between 9 and 14 pounds in weight. He could not tell from the external appearance whether they were male or female, but assumed that two thirds of his specimens were female, although he does not say why he thought so. Externally, there is little to distinguish male from female lungfish.

In O'Connor's early accounts of the work [14], he explains that of the 109 fish collected from the Mary River, 11 escaped and 12 died soon after capture. Nine expired during transit to their temporary home in a farm dam, where they were held for six months. Of the remaining 77 fish, 8 more died before the translocations began, leaving 69 fish, just over half of the number originally collected for the experiments. There were more losses during transport to their new homes. Eleven such deaths were recorded at the time. Therefore,

the translocations were based on the 57 known survivors, half of the animals originally collected from the source on the Mary River.

According to the account given by O'Connor in 1897, eight fish were taken to the North Pine River, well above the point where brackish water entered the river, on the 8 May 1895. Only three of these fish survived, just possibly too small a number to establish a viable population in the Pine River system. A recent analysis of mitochondrial DNA from Lake Samsonvale reveals a unique haplotype among the population, suggesting that lungfish were always in the Pine River system (Loh pers. com.). If the three Mary River fish survived in the Pine River, they may have only been added to an existing population.

On 17 November 1895, five fish were taken to a lagoon near the Albert River, south of Brisbane. Four of these survived. The Albert River is a small coastal river and not a good habitat for large fish. Few have been recorded from this river since the original introduction. Eight were taken to a farm dam near Cressbrook on the McConnell property near the Upper Brisbane River on the 15 December 1895 [14]. Of these eight fish, five survived the journey. The fish were not placed in the Brisbane River and may never have escaped from the farm dam.

The five fish from the Mary River that were taken to Cressbrook have been presumed to be the progenitors not only of the fish that formed part of a fish kill a few years later [15,16], but also of the entire present population of lungfish in the Brisbane River [17]. The dam in question communicates with the Brisbane River only during times of flood. Lungfish do move around in a river system, but fish, probably in poor condition, translocated to a farm dam are not in their own environment. Even if all five of the introduced lungfish from Miva were healthy when they were placed in the farm dam and survived there, and if a flood did cause the dam to communicate with the river, they may not have entered the river system. For the lungfish in question to have survived and entered the Brisbane River, produced enough descendants to be involved in the fish kill of 1900 or 1901 [15,16], and still leave progeny over for extending further into the river system, the five specimens would have had to have included mature adults of both sexes to have escaped quickly from the dam, settled in the river, spawned, and produced progeny. These would have had to become adult in their turn, all in the space of a few years. This sequence of events is not plausible.

Establishment of a population of lungfish in a new river system from only five fish is a little improbable, even if they did escape from the farm dam in reasonable condition. Although the growth rate of young lungfish in a good environment can be rapid, the ability of lungfish to recruit young to the adult population is not high in any of its known environments. Lungfish are dark in colour and benthic in habit, and when food is plentiful, are unlikely to take bait; therefore, they are hard to catch by conventional methods. Lungfish could have been living in the Brisbane River before O'Connor carried out his translocation experiments. Archaeological evidence indicates that they were [13].

O'Connor placed 18 fish in Enoggera Reservoir in May, 1896. It is not recorded if any of these fish died during transport. The translocation to Enoggera Reservoir was considered to have been unsuccessful because, at that time, none were seen again in the lake. O'Connor thought that his translocated fish in that site had all "been killed by shags, which infested the reservoir" at that time, and that they had eaten all the lungfish. Shags are not big enough to do any damage to an adult lungfish, and they have coexisted with lungfish for many years. In fact, lungfish settled quickly into Enoggera Reservoir and soon learned to use water hyacinth plants, introduced to the lake many years before, for spawning sites. Until water hyacinth was removed from Enoggera Reservoir in 1974, the lungfish population did well. However, without water hyacinth plants to use for spawning and for the young fish to use as refuges [20], spawning in this habitat ceased.

On the 31 July 1896, using water plants to enclose and protect the fish instead of wooden boxes containing only a little water, O'Connor took 21 fish to the Condamine River near Warwick. When rainfall is adequate, the Condamine is a big river and communicates eventually with the Murray/Darling catchment that passes through New South Wales and

Victoria, before joining the sea in South Australia. Lungfish are still occasionally caught in the Condamine or in one of its tributaries.

The last translocation was performed on the 29 August 1896, when 16 fish were taken to the Coomera River south of Brisbane. Two died during the night after they were released. However, 14 apparently equally sick fish survived and eventually moved further into the river [21]. Lungfish became established in the Coomera River and are still there, but not in large numbers. This river is really too small for such large fish and is now much affected by the suburban spread of Brisbane.

There has been at least one additional transfer of lungfish, later than those carried out by O'Connor. Thomas Bancroft, the son of Joseph Bancroft, placed a number of lungfish in Lake Manchester, in the hills to the west of Brisbane. This reservoir communicates with the Brisbane River at Kholo during times of heavy rainfall. These fish came from the Burnett River.

The translocation project must have been a logistical nightmare because travel in country areas of Queensland at that time was not particularly easy. A special grant from the Government, at the request of the Royal Society of Queensland, was provided for the work, and fish travelled freely on the fledgling railway system in the guard's van. However, a map of Queensland's railway network, published in 1926, shows that there was no rail link, even then, with Miva or Munna; thus, it must have been even more primitive when the translocation experiments were in progress. Nor were there railways to take the fish all the way to their new homes. It is not really surprising that so many lungfish died during the experiments.

### 3.5. The Diet of Lungfish

Analysis of gut and faecal contents of living lungfish from the Brisbane River and from Enoggera Reservoir indicates that they are suctorial feeders, often ingesting material that has no food value to the lungfish, as well as animals that can be digested and utilised [22]. Lungfish do not have the dental equipment of active carnivores, but they are very effective at drawing material into the oral cavity.

The intestines and faeces of adult lungfish from river environments include large quantities of plant and animal material, suggesting that the fish are omnivorous [23]. In lungfish from the Brisbane River, the plant material consists mostly of filamentous algae, *Rhizoclonium*, with occasional fragments of *Myriophyllum*, none of which are affected by chewing or by digestion. Broken shells of snails, *Thiara* (*Plotiopsis*) *balonnensis*, and small clams, *Corbicula australis*, are most common among the animal residues in the intestines, with the occasional scale from a small fish and fragments of a shrimp carapace. Faecal samples contained a similar mixture of filamentous algae, mollusc shells, and a few other animal fragments. Plant material in the faeces has not been altered in any way and can still be identified. This indicates that the fish in river environments obtain their nutrients almost entirely from snails and clams. Significant amounts of sand or gravel are also ingested, but are removed in the faeces.

Lungfish in Enoggera Reservoir had a restricted and harsh diet. Intestinal and faecal samples included leaves of *Hydrilla verticillata* and the broken shells of small freshwater snails, with a few fragments of shrimp carapaces, as well as rootlets of the water hyacinth, *Eichornia crassipes*, ingested as the fish search for food among the pendulous root masses. Plant material is not digested. Food in this reservoir is not abundant, nor is it varied [7]. Most of the plants that are found in Enoggera Reservoir occur around the margin of the lake. These are mainly water hyacinth, water lilies, para grass, and *Myriophyllum*, none of which appeared in the intestinal contents or faeces. Filamentous algae are not common in this environment and are not found in the intestines or faeces of fish from this habitat. A few submerged water plants, such as *Hydrilla verticillata*, small water snails and freshwater prawns, such as *Paratya*, are present in the lake, and traces of these food items are found in the intestines and faeces of lungfish from this locality [22]. Most of the plants found in

Enoggera Reservoir are not eaten by lungfish, and the animals that are part of the diet of large fish like *N. forsteri* are found only around the shore.

Although there is little published information available on the food available in fossil habitats or on the dietary preferences of the fossil species, the structure of the dentition and wear characters on the tooth plates indicate that fossil species ate similar materials as the living species and suffered from similar disease conditions.

### 3.6. Tertiary Fossil Populations

Tertiary lungfish are mostly classified in *Mioceratodus*, and a few belong in the related genera of *Archaeoceratodus* or *Metaceratodus* [1,2]. Species of *Neoceratodus* are found in fossil environments [24], but are not common as fossils, and *N. forsteri* is not represented until the Pliocene [25]. Lungfish belonging to these genera were widespread in central, northern, and eastern Australia, but a gradual contraction towards coastal rivers is evident.

### 3.7. Localities of Known Provenance

3.7.1. Palaeocene, 65–56 Mya

Redbank Plains in the Redbank Plains Formation, Queensland, Eastern Australia

Palaeocene records of lungfish come from Redbank Plains (Figure 5) near Brisbane [26,27]. The locality, based on a shallow lake and the only Tertiary site that is near the coast, has several species of fossil lungfish. The most complete specimen is an impression of the skull bones and partial dentition, originally described as *Epiceratodus denticulatus* [28]. The material has been reassigned to *Mioceratodus gregoryi*, based on the morphology of the dentition [1,29]. The species is common elsewhere later in the Tertiary. The specimen has a developmental anomaly, with unusually short ridges in one prearticular tooth plate. The dentition shows no sign of disease or abnormal wear, and the specimen is within the size of a subadult fish. It may have died when water levels in the lake fell [29]. Several additional tooth plate fragments have been collected from Redbank Plains and have been assigned to *Archaeoceratodus djelleh*, also found in Central Australia. These appear healthy, with normal wear. A number of larger tooth plates from adult fish have also been collected from this site, but have not yet been described (Rix, pers. com.)

The impression of a large lungfish tail was included in the first description of fossils from the Redbank Plains site [28]. However, a tail is not diagnostic at a species level and provides no information regarding the diet or the health of the fish.

3.7.2. Eocene, 56–34 Mya

The Rundle Formation in the Shale Oil site at Rundle in Queensland (Figure 5), has one lungfish. *Archaeoceratodus rowleyi* is represented by one nearly complete tooth plate and one fragment [1]. Both specimens are devoid of disease conditions and represent adult fish. Wear characters suggest normal grinding usage of the dentition.

3.7.3. Oligocene, 34–23 Mya

The central Australian environments of Oligocene times can be arranged in several groups (Figure 5), possibly of similar age, but with different characteristics. The Namba and Etadunna Formations are based on large lakes, and the Wipijiri Formation is derived from a river [30]. All of these areas had large numbers of lungfish.

At the peak of the Oligocene lungfish radiation in Central Australia, there were at least eleven fossil species, all represented almost entirely by tooth plates, but with a few skull bones [1,2]. They ranged in size from the gigantic specimens of *Mioceratodus*, which may have reached a length of several metres, to small and insignificant specimens related to the living Australian lungfish, *Neoceratodus*, and species of *Archaeoceratodus*. Large adult lungfish dominated the fauna, grew to a great age and massive size, and mostly disappeared as the continent became more arid and the lakes disappeared.

Little is known of the plants and animals that made up the diet of the Oligocene fossil species in central Australia, although the structure of the dentition and the way that the

tooth plate wore suggests that food ingested would have been similar to that of the living lungfish [22]. The fossil environments had gastropods, *Cladocera*, and probably other plants and invertebrates [31,32]. Many of these lungfish could grow very large, but they certainly did not possess the dental equipment and jaw structure of an active carnivore. The little that we know of their morphology, including the hyoid apparatus that supported the tongue [29], indicates that, like the living species, they made use of suctorial feeding and either crushed or ground the hard items that they ingested. The mid-Tertiary environments of central Australia would have been more extensive and diverse than the later deposits in northern Australia in Miocene times, but perhaps not always what the lungfish of those times actually required.

### 3.7.4. Namba Formation

The deposits of the Namba Formation (Figure 5) formed from an extensive system of lakes, rivers, and perhaps an estuarine environment created during times of high rainfall, but with seasonal periods of drought [31]. The environments of the Namba Formation could have been, at times, brackish or even saline.

Most of the species of lungfish present in this Formation have been assigned to species of *Mioceratodus*, mostly *M. gregoryi*, *M. poastrus*, or *M. anemosyrus*. Some are *Neoceratodus eyrensis*, and a few are species of *Archaeoceratodus*, including *A. djelleh* and *A. theganus*. Most of the lungfish of the Namba Formation were exceptionally large. Wear on the tooth plates was often heavy, and examples of both crushing and grinding usage of the tooth plates are present. Attrition affects nearly a quarter of the tooth plates, suggesting that food was often absent [7].

Lungfish were common in these localities, although the Namba Formation has no small specimens. This indicates that recruitment had ceased, perhaps because of few suitable spawning sites in the lakes or a lack of juvenile habitats. Although the lungfish undoubtedly lived to a great age, they were not particularly healthy. Most tooth plates from this deposit, whatever the size, are heavily worn, although the incidence of spur and step wear is low, and about one third used crushing movements of the jaws. Although malocclusion, trauma, and hyperplasia are often present, evidence of erosion and caries are infrequent, and only one specimen with serious disease-related pathology was collected.

### 3.7.5. Etadunna Formation

The Etadunna Formation, contemporaneous with the Namba Formation and with a similar species composition, mostly of species of *Mioceratodus*, arose from large permanent freshwater lakes. Fossils of wetland birds, such as flamingos and ducks, indicate that the lakes may have been alkaline, surrounded by woodlands, and maintained by freshwater streams [30]. The Etadunna Formation has a few juvenile tooth plates, some still having cusps on the occlusal surface, indicating that they are very young. Although recruitment was occurring among the lungfish of this Formation, many of the tooth plates, including a few in the smallest size class, are heavily worn, and nearly half were used for crushing abrasion. Spur and step formation is present in many tooth plates. The incidence of erosion is higher than in any other locality, and attrition affects over one fifth of the specimens. Among fossil deposits, caries is also high, but pathology, apart from malocclusion and hyperplasia, is rare [7]. This was an environment with many large and old fish, some juveniles, plentiful but probably harsh food, and water that was acidic at times, as in the Namba Formation deposits. There were some small tooth plates, suggesting at least some active recruitment, but pathologies and wear indicative of an inadequate diet are common in tooth plates from this locality. It is also the only fossil site to have preserved an example of a tooth plate with damage from a nematode worm, as has been described in one living Australian lungfish from the Brisbane River [7].

### 3.7.6. Wipijiri Formation

The third mid-Tertiary deposit in central Australia, slightly younger in age than the Namba and Etadunna Formations, is the Wipijiri Formation (Figure 5). This habitat may have been formed by the intrusion of a freshwater river into the Etadunna deposit. The fossils in this locality differ in significant details from those of the Namba and Etadunna Formations. The Wipijiri Formation was based on a rich environment and had large numbers of lungfish, most of them belonging to the commonest of the mid-Tertiary taxa, *Mioceratodus anemosyrus* [1]. Recruitment was extremely active in this deposit, and many of the tooth plates belong to small, young fish. Also notable was the lack of pathologies, and the tooth plates had smooth wear. This indicates that the environment was based on a riverine locality, similar to the Brisbane River up until 1990, with plenty of food [7]. The lungfish were healthy, and it was not old age and disease that brought the population to an end.

Lungfish from the Wipajiri Formation were spawning successfully, with sizeable recruitment of young to the adult population. Few exceptionally large specimens are present in the Wipajiri Formation, and the fish did not attain such a large size as they do in other Oligocene environments. The incidence of spur and step wear is low. Environmental quality in the Wipijiri Formation was good, and tooth plates have a low incidence of caries. The water was not acidic, and severe erosion was infrequent. Pathologies suggestive of a poor environment are absent. Lack of attrition among the tooth plates and growth to a reasonable size indicates that strata of the Wipijiri Formation represent rich environments with plentiful food.

### 3.7.7. Miocene, 23–5.3 Mya
#### Northern Australia

Lungfish of Bullock Creek and Riversleigh in the north of Australia belong in Miocene deposits that are younger than the central Australian deposits. They were based on rivers, possibly running through grasslands or a forest, similar to the modern Gregory River in far north Queensland. These localities have only six different species of lungfish, most belonging to *Mioceratodus*, and all of these species were also found in central Australia.

### 3.7.8. Camfield Beds

The Camfield Beds in the Northern Territory, at Bullock Creek, developed from a paleochannel that included slowly moving rivers and pools with permanent water, sometimes quite deep, with shallower margins and levels that varied with the seasons (Figure 5). Sedges and grasslands, but not forests, may have surrounded the wetlands [33]. The conditions around the wetlands, beyond the grasslands, may have been arid, as they were elsewhere in northern Australia and still are at any distance from permanent water. This palaeochannel had a large population of rather small lungfish, and the freshwater turtles that lived here were also small [34], suggesting that conditions were not conducive for the fish or the turtles to reach a large size, perhaps because of crowding or limited food.

Most of the Bullock Creek fossils consist of tooth plates and attached bones of *Mioceratodus anemosyrus*, and these are well preserved. The lungfish from this area at that time are unlikely to have come from old fish. Not one of the tooth plates has come from a large fish, although the species to be found at Bullock Creek could grow to a large size, and did in other parts of Australia in older deposits. Two thirds of the tooth plates come from fish of medium size, up to 80 cm in length, and the rest are less than 40 cm long [7]. Although small, many tooth plates show signs of heavy wear and evidence of crushing jaw movements, suggestive of a harsh diet. Caries is present in about a quarter of the specimens. A few have attrition or trauma, and many of the specimens in the medium size class have spurs on the posterior heel of the tooth plate, a sign of age. Although pathological specimens in this environment are rare, apart from caries, and young fish were being recruited to the adult population, they did not grow to a large size. Perhaps food or living space were limiting in some way.

### 3.7.9. Carl Creek Limestones

The environment of Riversleigh (Figure 5) in far north Queensland had some similarities to the wetlands that ultimately produced the Bullock Creek sediments. The deposits that created the Carl Creek Limestones of Riversleigh may have been based on perennial freshwater streams with deep pools, filled by springs and surrounded by rainforest elements and, further away from the water, by arid country, much like the modern Gregory River and its surroundings now [35]. The Gregory River is still there, a large flowing stream with many freshwater fish and reptiles, fringed by a narrow band of thick rainforest. There are no lungfish in this river at the present time.

The limestones in the fossil localities demonstrate that the waters that formed the Riversleigh deposits were high in carbonate [35], and although the rainforests along the margins of the river and the deep pools may have provided a rich environment for terrestrial wildlife, the waters of the nearby river most certainly did not. The large numbers of lungfish, mostly assigned to *M. anemosyrus*, in the Carl Creek Limestones are, like the turtles, small, with the single exception of one large tooth plate of *Neoceratodus eyrensis*, rare in most of the fossil environments. The species composition of the various localities in the Carl Creek Limestones is similar. The deposits of the Carl Creek Limestones supported populations of lungfish that were able to spawn and recruit young to the adult population, but few of the fish were able to grow to a size commensurate with that of the same species in other localities in Australia in the Oligocene [7].

Preservation of material from the Carl Creek Limestones is less perfect than it is at Bullock Creek, and no skull bones have been recovered from Riversleigh. Spur and step wear on the tooth plates is not common, and only half of the specimens show signs of heavy wear, with even fewer using crushing movements indicative of a harsh diet [7]. Erosion is unusual; thus, the water in which the fish lived would not have been acidic, but caries, malocclusion, trauma, and hyperplasia of dental tissues are common, all indicating damage to the tooth plates from rough items in the diet. Attrition and pathologies indicative of a poor environment are not particularly common, but other characteristics of the tooth plates suggest that the environment may have been in some ways depauperate, and the food, even if it was plentiful, was not particularly nutritious. Large numbers of small specimens also suggest crowding or some other factor that limited the size of the adult fish.

### 3.7.10. Pliocene, 5.3–2.6 Mya

Katapiri Formation

The numbers of different species of lungfish in Pliocene deposits in central Australia are severely limited, with only a few of the species found in Oligocene deposits still present in the two remaining central Australian deposits of Pliocene age.

The Lower Cooper Creek in Central Australia (Figure 5), a correlate of the Katapiri Formation elsewhere in South Australia, contains several species of lungfish [36]. This is the type of locality of *M. gregoryi* and *N. eyrensis*, as well as *Mioceratodus diaphorus*. Lungfish in this deposit were not particularly large and had few pathologies, although some tooth plates came from old fish. There are no small specimens in Lower Cooper Creek [1].

The Katapiri Formation in central Australia (Figure 5) was based on Lake Kanunka, which was a huge lake, mostly deep and possibly saline, surrounded by meandering rivers and having many shallow areas. All parts of the environment may have been affected by fluctuating water levels that prevented the growth and establishment of submerged aquatic plants around the margins of the lake, or allowed much to grow on the shores. Lungfish of the Katapiri Formation represent fewer species of *Mioceratodus* than in the older Central Australian environments, and these were all from large fish. The tooth plates are all heavily worn, eroded, and carious. Many large and diseased tooth plates, and no small specimens, suggests a dying population with little or no recruitment.

Eastern Australia (Catchment of the Condamine River)

There is a single eastern Australian deposit that can be dated reliably to the Pliocene, and the environment in this place was richer and more suitable for lungfish than the Katapiri Formation (Figure 5).

Chinchilla Local Fauna (Early to Middle Pliocene)

Lungfish in the Chinchilla deposits, in Queensland west of the Great Dividing Range, belonged to two species, *Neoceratodus forsteri* and *Metaceratodus palmeri* [1,2]. The latter grew to a large size, although the *N. forsteri* material was commensurate with the sizes of living *N. forsteri*. All of the tooth plates from this deposit were smoothly worn with shallow furrows and rounded crests, suggesting a plentiful and soft diet, similar to the fish of the modern Brisbane River until 1990. Few show signs of disease, although erosion of the mediolingual face occurs in larger specimens of *M. palmeri*. However, there were no small tooth plates of either species found in Chinchilla, thus it is unlikely that young were being recruited to the adult population.

3.7.11. Pleistocene Deposits, 2.6 Mya—10,000

King's Creek, Eastern Australia

*Metaceratodus palmeri* was found in Pleistocene deposits in Queensland (Figure 5), within the catchment of the Condamine River, and the fish reached a large size [2]. There are no small specimens of this lungfish in the King's Creek locality, and it is likely that it is a relict population, although the lungfish were apparently healthy.

The living Australian lungfish is present in the Condamine catchment, a tributary of the Murray/Darling Rivers, but only because it was translocated from the Mary River [14] during the efforts to preserve the lungfish from extinction.

*3.8. Quaternary Lungfish*

A single specimen of *N. forsteri*, from the time of the megafauna, has recently been collected from Toorale Station, on the banks of the Darling River, a tributary of the Murray River (Figure 5). The megafauna bones found with the specimen have been dated to 170 kya. The tooth plate is well preserved and free of disease conditions. Lungfish tooth plates are apparently common from this locality (Westaway pers. com.). There are no living lungfish in the Darling River or in the Murray River now.

Moorna Formation, Southern Australia

Lungfish have been found in the Moorna Formation at Chowilla, late Pliocene or early Pleistocene in age (Chowilla Sands, Bone Gulch Fauna). This deposit is based on an extensive floodplain within the catchment of the Murray River in South Australia and extends into New South Wales (Figure 5). The material consists of a handful of fossils belonging to three species, *Neoceratodus forsteri*, *Mioceratodus gregoryi*, and *Metaceratodus palmeri*. All of the specimens came from old fish, rather like the specimens found in the Katapiri Sands, or in Enoggera Reservoir in recent times. The tooth plates were eroded, diseased, and broken, with large carious lesions and heavy crushing wear. The bones of the jaws were affected by osteoporosis and very fragile. There are no small young tooth plates. This was not a healthy population, and it is no surprise that it became extinct.

*3.9. Lungfish in Deposits of Uncertain Provenance*

One lungfish, *Archeoceratodus djelleh*, has been described from the Duaringa Basin in central Queensland (Figure 5) [37]. The specimen is well preserved and was heavily worn during life. This species also occurs in Oligocene localities in central Australia, but is never common. This deposit cannot be dated with any certainty as pollen is absent.

There is one record of a fossil lungfish, supposed to be of "*Ceratodus forsteri*", a tooth plate found, along with the jaw of a fossil lizard, in a well on a property at Eight Mile Plains, now part of the city of Brisbane [38]. The specimen is still in the collections of the

Queensland Museum. The title of the paper reads "*Ceratodus forsteri* post Pliocene" and records the presence of the tooth plate within the Brisbane River catchment, but not far from the coast, where the river was brackish or actually tidal. "Post Pliocene" means that the age and the source of the fossil are uncertain. The tooth plate (Figure 2B) is a pretty specimen, preserved with many details of value to the taxonomist, such as a particular pattern of punctations on the occlusal surface and short parallel ridges. There are no pathological conditions on the tooth plate. It shows crushing abrasion.

It is not a specimen of *Neoceratodus forsteri*, but belongs to the related fossil species, *Metaceratodus palmeri* [2], known from Pleistocene and Pliocene deposits at Chinchilla and Kings Creek on the western side of the Great Dividing Range. The specimen may have been reworked, found elsewhere, and deposited in the well at some later date. Even if the fossil did originate from Eight Mile Plains, this record of "*Ceratodus forsteri* post Pliocene" does not represent an occurrence of the modern *Neoceratodus* in the Brisbane River system prior to the activities of O'Connor in 1897.

*3.10. The Condition of Living Lungfish*

All of the coastal rivers that provide lungfish habitats are now affected by water impoundments, such as Lake Wivenhoe and Lake Somerset in the Brisbane River catchment, Lungfish from the Brisbane River, below the wall of Lake Wivenhoe, collected before the river was altered by drought and flooding were in reasonable condition. There was little sign of poor health, and the colour of the fish, dark brown on the dorsal surface and pink on the belly, was normal. The scale cover was complete and undamaged. The gut and faecal contents indicated that the fish were feeding well. Observations of the river environment indicate that both plant and animal food was plentiful in the river at that time.

Many areas of the rivers had extensive beds of eel grass, *Vallisneria spiralis*, in the shallows where small molluscs, such as basket clams, *Corbicula australis*, lived. In deeper water, there were aquatic snails, such as *Thiara balonnensis*, on rocks and among weeds, and large bottlebrush trees with submerged roots hanging down into the water, providing a home for basket clams, rotifers, worms, and small prawns. The river was free of weed species, such as cyanobacteria, water hyacinth, and para grass. The water was clear and fresh. Eel grass is one of the water plants favoured by lungfish for spawning sites, but it is not the only plant used for spawning. Lungfish search for this plant because food animals, such as snails and clams, live amongst the leaves, but if they ingest any eel grass by accident, it is not digested.

Tooth plates of lungfish from the Brisbane River at this time had a smooth occlusal surface, rounded ridge crests, and shallow rounded furrows in the clefts between the ridges [7]. There is little or no incision of the ridges on the tooth plate, indicating light grinding mastication of food items enough to break the shells of snails or clams and prepare the ingested animals for digestion in the anterior sac and spiral valve of the intestine. One specimen had a small exostosis on the medial face, and two specimens have malocclusion. Spur and step wear is present in 8 out of 11 specimens, and 2 have osteopenia, 1 quite severe. All have mild erosion of the mediolingual face, and 5 out of 11 specimens and Lake Samsonvale in the Pine River, (Figure 4) as well as Paradise Dam on the Burnett River. These reservoirs do not support the production of viable young among the trapped adult population. Food and refuges for hatchling and juvenile lungfish are effectively missing in these water impoundments, and all of the eggs spawned in recent years in these habitats have died [39–41]. Most of the surviving wild populations of the lungfish in these habitats consist of large adult fish, and subadult and juvenile fish are rare or absent [42,43].

Spawning ceased in parts of the Brisbane River below Lake Wivenhoe during the drought of 2001–2008. The environment was affected by the subsequent flooding in the river, and it is unlikely that spawning has occurred in the river in recent years. Spawning has occurred in water impoundments [44]. However, embryos collected at that time in Lake Wivenhoe, and later in Lake Samsonvale, and those taken for raising in the laboratory were

too abnormal to develop, and all died. Recruitment will not have followed the spawning events [39–41].

### 3.11. Lungfish in the Brisbane River Catchment before 2001

Have small carious lesions. Hyperplasia is absent. The size and condition of the tooth plates indicates that the dental material came from large adult fish, and some of the changes, such as erosion, osteopenia, and spur and step wear, suggest advanced age. One specimen was infected with nematode worms in both tooth plates and in the bones of the jaws and skull.

Small tooth plates were also present among river fish at this time, and the population was actively spawning [45]. One of the juvenile tooth plates had been broken, and this had caused abnormal growth in the opposing tooth plate. The other small tooth plates in the collection showed little pathology, and wear was suggestive of light grinding usage.

### 3.12. Lungfish in the Brisbane River Catchment after 2001

Spawning came to an end in the Brisbane River below the wall of Lake Wivenhoe in 2003 [45], during a severe drought. Food availability in the river, for adults and young fish, was limited at this time, and one fish found dead on the river bank in 2005 had an empty intestine. The drought ended in 2008 and subsequent rain was heavy, beginning in 2009 and culminating in a massive flood of 2011, after which significant rain continued for several years. The flooding of 2011 removed plants and animals from the river, reduced food for all life stages of the lungfish to very low levels, and destroyed spawning sites and refuges for young fish completely. The river below the wall of Lake Wivenhoe has not recovered from the flooding of 2011 [5,46] and has recently been badly damaged again by a flood in February 2022.

A number of adult lungfish, from an extensive fish kill at the head of Lake Wivenhoe, were collected after heavy rain in July 2009. They were trying to swim upstream during the flood, were trapped, and died among rocks when water levels began to recede. The fish appeared to be in good condition on external inspection, but the intestines were empty. Wear on the tooth plates suggests heavy usage with attrition, indicating that the fish had no food for a long time [5].

### 3.13. Lake Samsonvale Lungfish

Eight fish, also adults, were collected from the area below the wall of the dam at Lake Samsonvale (Pine River catchment) after a flood event in 2009, in early summer, when they were washed over the wall of the reservoir [5]. Most of these were left exposed on the bank when the floods receded. The body condition of these fish was poor, and the fish were thin. The dorsal surface of the fish was a dull brown and the belly a dull pale yellow, colours that indicate poor health in an adult lungfish. These colours are characteristic of lungfish from Lake Samsonvale.

Post mortem examination of nine fish from the Lake Samsonvale spillway pool revealed traces of undigested filamentous algae in the rectum. Several fish had fragments of tree leaves in the posterior intestine. Filamentous algae and leaves are the only items available to be eaten in this habitat, and these items are not digested [5]. The anterior sacs and the anterior intestine of all the fish were empty. The fish kill in this system occurred in early summer, when food items are present in most environments, but not in spillway pools below reservoirs.

Medial erosion with caries is present in most of the tooth plates. Three out of five specimens show spur and step wear, and one has malocclusion in one jaw. Most of the tooth plates show attrition, indicating that the fish had not been chewing any food [5]. Recruitment of hatchings to the Lake Samsonvale population has ceased, and the adults have not spawned since 2018 [6,40,41].

### 3.14. One Documented Extinction of the Extant Lungfish

Enoggera Reservoir was the first water impoundment established in the city of Brisbane. It was built in the hills to the west of Brisbane in 1867 and is surrounded by dry sclerophyll eucalypt forest. Because the surrounding hills were so steep, Enoggera Reservoir is deep, with sharply sloping banks. Water plants and animals that need to shelter in them can only live in the narrow margins of the reservoir and in the remnants of the creek that enters the lake. Over the years, detritus has gathered in the deeper parts of the lake, and few animals or plants can live in such an anoxic environment.

The land around Enoggera Reservoir was not cultivated, except for a brief period after the Second World War when returned servicemen were provided with plots of land in the hills close to the reservoir. Land surrounding the reservoir has grassy slopes with some large trees that are the remains of rainforests that formed the dominant vegetation of the valley in which the reservoir was created. The creek entering the reservoir, although quite large, never communicated with freshwater regions of the Brisbane River. The creek leaving the reservoir joins the Brisbane River as Breakfast Creek, well below the tidal reaches of the river. Enoggera Reservoir is isolated by position and by distance from any catchment containing lungfish and certainly had no native population of this species.

In 1866, within a year of its introduction to a pool in the gardens of the Queensland Museum, an introduced plant from South America, *Eichornia crassipes*, the water hyacinth, made its way to Enoggera Reservoir. These floating plants soon colonised the whole lake. The plants were large, with long dense trailing roots, and formed thick mats in shallow water along the shore. In spring, it was confined to the margin of the reservoir, but filled more of the surface of the lake as summer progressed. Although now a despised and noxious invasive weed, the hyacinth provided a perfect habitat for snails and small prawns, as well as other small invertebrates. Enoggera Reservoir is not a rich environment, and the species diversity of plants and animals living in the lake may well have been improved by the presence of hyacinth plants. When lungfish were translocated into the reservoir environment in 1896, they used the water hyacinth for spawning and searched for food among the submerged roots.

This idyllic situation changed early in 1974. Enoggera Reservoir, at that time retained by the original low earth dam wall, contributed to the flooding of Brisbane suburbs, caused by exceptionally heavy rain in 1973. Water hyacinth was carried out of the reservoir and into the creek, along with many fish. The water hyacinth choked the creek. After the flood, the water hyacinth was poisoned with herbicides, thus destroying refuges for eggs and young hatchlings. With no protected spawning habitat, no embryos and young lungfish survived, and the ageing adults of the reservoir have become extinct. A few individuals have apparently survived in Enoggera Creek, but this is a poor environment for large fish.

Clearing of the hyacinth after the floods of 1973 was not the first time that this plant had been removed from the reservoir. A few years after juvenile lungfish were found hiding in water hyacinth roots in Enoggera Reservoir [20], the weeds were cleared by forking the hyacinth plants onto the bank. At the time, people asked for hyacinth to be allowed to grow back in the reservoir to protect small animals, and this happened, probably because nature took a hand in the matter. The hyacinth may not have been completely cleared and was able to grow back. Clearing of the hyacinth was a lot more efficient in 1974, using weed killers and applied by a retired farmer who was curator of the reservoir at the time.

The condition of the lungfish of Enoggera Reservoir, after the water hyacinth was cleared from the reservoir in 1974 and before the population became extinct, was poor. Analysis of a series of matched tooth plates from this locality, collected between 1981 and 1988, after recruitment ceased in the reservoir, tells a very different story compared to the Brisbane River fish of a similar time [7]. There are no small tooth plates, no juveniles, among this collection. They have all come from large adult fish. Recruitment of young animals to this population must have ceased in 1974 when the water hyacinth was cleared and the adults left behind were getting older and older. The contrast with material from the

Brisbane River collected in 1990 before the long drought (2001–2008) and the subsequent flooding is frightening.

All of the tooth plates from Enoggera Reservoir, collected after the water hyacinth was cleared and spawning ceased, came from large adult fish. The occlusal surface is incised from the labial to the medial margin in every specimen. Slight hyperplasia of individual ridge crests occurred in nine of the specimens, with a correspondingly deep furrow in the opposing tooth plate. One specimen shows extreme hyperplasia (Figure 3). Erosion of dental tissue, usually with deeper carious lesions exposing the pulp cavity, was found in every single tooth plate along the medial face. One has small traumatic lesions on the labial face of every ridge and others have similar lesions on single ridges. Seven of the specimens showed spur and step wear, and the attached bones of the jaws of every tooth plate were affected by osteopenia. All of the tooth plates were severely worn, with high, faceted ridge crests separated by deep furrows. Wear on every tooth plate from Enoggera Reservoir is extreme, and there are no juvenile tooth plates at all. There are now no suitable water plants to act as spawning sites in Enoggera Reservoir, and no refuges for young fish.

The lungfish of Enoggera Reservoir, all adults at the time of translocation in 1895, senescent in 1981–1989, and now extinct, provide an upper limit for the age of wild lungfish of at least 110 years old, plus the age of any translocated fish. It is unlikely that they can live for so long in the wild, perhaps 70 to 80 years at the most [46]. Lungfish spawned freely in Enoggera Reservoir for many years, and some of the fish caught after 1974 must have been the progeny of the original translocated fish. The lungfish of Enoggera Reservoir also suggest a lower limit to the possible age because successful recruitment stopped in 1974. The youngest lungfish caught in Enoggera, if there are any still there, would be at least 48 years old now. However, every lungfish caught in Enoggera or dying of an accident in the creek from 1980 till 1993 was an old fish, with heavily worn carious tooth plates, osteoporotic bones, and a heavy load of parasites. At that time, none of the fish collected were as young as 10 or 20 years, as they would have been if derived from eggs laid after the water hyacinth was cleared. They were all much older. In other words, it took around 40 years from the time spawning sites were destroyed to the time that extinction of the adults happened, a fact that was admitted eventually in public by Government officials.

## 4. Discussion

The last remaining member of the Neoceratodontidae, the living Australian lungfish, *N. forsteri*, is now confined to coastal rivers and reservoirs in southeast Queensland, and the species is under threat from environmental degradation, which has removed food for adults and young fish and spawning sites and refuges for hatchlings [4]. Rivers have changed fundamentally since the lungfish first evolved and since the lungfish became isolated in coastal river systems.

Before European occupation, rivers flowed freely, with no dams and no weirs. Now, the few remaining rivers where lungfish have survived are blocked by numerous reservoirs, and little is done to protect or restore the environment in the reservoirs or the remaining rivers after adverse weather events. This is not to say that many thousands of years of periodic droughts and floods before reservoirs were built never had any effect, but at least the rivers could recover quickly after such catastrophic events, and lungfish could survive. This is no longer the case.

Similar issues appear to have affected some of the fossil populations. Early in the Tertiary, suitable environments for the various species of lungfish occurred in many places in central and eastern Australia. Although the state of lungfish in any of the fossil deposits is not the same as it is now, it is possible to draw a few conclusions. Extinction of the central Australian taxa did not simply arise from increasing aridity of the environments in many of the areas where lungfish lived. Perhaps the lungfish of the river environment of the Wipijiri Formation died out when the river dried up, or the periodic loss of the lakes that formed the Namba and Etadunna Formations made life for large freshwater fish untenable. The actively spawning population of Wipijiri fish conform in dental structure and composition

of the lungfish fauna to the material found in the Brisbane River up to the time of the long drought, but have far fewer instances of disease or trauma to the tooth plates and much larger numbers of small specimens, suggesting a high recruitment rate of young to the adult population. Lungfish of the Namba and Etadunna Formations, based on lakes, were mostly old and in a poor environment with few juveniles, and they may have died out because recruitment failed or because the environments became unsuitable. The population of the Katapiri Formation in Lake Kanunka certainly died out from lack of recruitment and increasing disease of the remaining adult lungfish, and the environment was poor as well, similar to the situation in Enoggera Reservoir before the lungfish became extinct there. In the mid Quaternary, we can tell from fossils that whole populations, such as the lungfish found in the Murray River, stopped producing any young and became older and sicker before they disappeared, as in Enoggera Reservoir now. This happened although some of the rivers and lakes they lived in were still there, and, in the case of the Murray River and the Gregory River in the north, are still there.

The habitats in which lungfish are endemic are not as seriously limited as the authorities insist [17,18], because lungfish are endemic to the Brisbane River, as well as the Mary and the Burnett Rivers, although the distribution is certainly restricted. The members of the Royal Society who became interested in the fate of lungfish in 1895 were not field biologists and did not realise that the Brisbane River had its own population of lungfish. They were scientists in other fields who were interested in lungfish. They lived in Brisbane, and, to be quite honest, they probably never looked for lungfish where they could be found, a long way from the city. The catchment of the Brisbane River is vast (Figure 4), and travel was not easy in those days. The "only found in the Mary and Burnett" definition was probably accepted without question, as it still is, by the way, with much less justification. As my old supervisor used to say, the whole story reeks of ecological vandalism. They simply did not know what the distribution of lungfish actually was when they transported them around the state. It was, after all, only about 25 years after the first records of lungfish were published.

The early scientists also believed that the species was about to become extinct, but somehow a large number of lungfish were collected from a small area of the Mary River in a fairly short time. Many of these fish died between the time of initial capture and eventual translocation. The admission that many of the lungfish caught for translocation died during the course of the experiment [14] raises two questions. Nearly half of the fish died during the early stages of the work, suggesting that they were not healthy, either because of the conditions under which they were kept since capture at Miva or because of injuries sustained during capture or transit. The lungfish populations in three of the rivers, if they arose from translocated fish, were established on very small numbers of animals, possibly consisting of males only or females only, and were not in fact successful. If large numbers of lungfish were found subsequently in the river system or lake, as in the Pine River, the translocated fish may simply have been added to an existing population.

Unfortunately, the ecological vandalism implicit in the translocation experiments means that the original distribution of the lungfish is not known and can now never be determined with any certainty. Government agencies have been able to make out that the Mary and Burnett are the only original habitats. They are most unwilling to give up entrenched ideas. The translocation experiments carried out in 1896 by O'Connor to "preserve our interesting Dipnoi from extinction" were misguided, to say the least, and carried out before the actual distribution of the lungfish had been determined.

Now, the riverine habitats of the lungfish are further constrained by numerous reservoirs, where lungfish are trapped unless they can safely escape during a flood. Very few of these reservoirs have fishways, such as Lake Wivenhoe and Lake Somerset on the Brisbane River and Lake Samsonvale on the Pine River. These water impoundments have high walls, and an undisclosed number of lungfish were carried over by recent flooding and killed. This happens during every flood. Lake Wivenhoe and Lake Samsonvale have gates that can be opened to release water during a flood. Lake Somerset has valves to release excess water, but if there is too much for the valves to cope with, the excess is carried over the top

of the dam wall. Fish and other animals sense the rapid flow of water and are swept over the wall or through the gaps. On Paradise Dam on the Burnett River, there is a narrow slot, known as the downstream fishway, covered with water during a flood, but no competition for the massive and rapid flow over the wall of the dam, even if it was wide enough to permit the passage of a large fish such as an adult lungfish.

The environments of several water impoundments in southeast Queensland, specifically Lake Samsonvale on the North Pine River and Lake Wivenhoe, Lake Somerset, and Mount Crosby Weir on the Brisbane River, including the wildlife and plants, are not covered by the environmental protection laws of 1992 (Queensland), 1994 (Queensland), and 1999 (Federal) because they were built before these laws came into being. The same applies to the 11 weirs across the Mary River, and presumably to the older weirs on the Burnett River. Collectively, these weirs and reservoirs take up or affect a significant proportion of the natural lungfish habitat, especially in the Brisbane, Pine, and Mary Rivers. Now, when the authorities build high walls on either side of the spillway and close off public access to the river or creek downstream of the spillway after a flood in any of these places that are not covered by the more recent legislation, and they remove or bury dead animals and rescue any live fish, they are simply looking after the environments under their care, as they should. They are under no legal obligation to do this job, and the numbers of endangered animals killed or injured are kept secret. Before the recent flooding of 2011, local farmers, conservationists, schoolboys, or angling societies always performed the rescues and were happy to tell me what they had done, although nothing was ever published on the subject. Spillways below the walls of the dams and reservoirs are now closed to the public, and the authorities do not release any information on the numbers of lungfish that are rescued or that die when they fall into the spillway.

SEQwater, the statutory water authority that has responsibility for the reservoirs of southeast Queensland, monitors lungfish populations in the area and has asked me to include the following statement in any of my publications that refer to the continuing plight of the extant lungfish.

"In response to the 2009 accidental lungfish deaths in southeast Queensland dams, SEQWater developed the first Lungfish Management Program by a water authority in Australia. SEQWater prepared a detailed post flood response plan for all three gated dam sites (Wivenhoe, Somerset, North Pine) that requires, amongst many things, rangers to undertake detailed inspections of spillways immediately after flood gate operations cease, recover any stranded lungfish and place them in permanent water. In addition, they record any accidental fish deaths. SEQWater has also tested a range of post flood release strategies to reduce the risk of fish stranding at spillways. They have also implemented improved communication processes to alert in advance of potential fish stranding incidents, recording of incidents and reporting to the appropriate authorities when required. Extensive civil works in high risk areas below dams to reduce the risk of fish damage and stranding during and post flood releases have been undertaken, and specialised equipment to assist in large scale lungfish recovery and research efforts has been purchased."

It is not known how effective this elaborate plan actually is. Certainly, numerous lungfish are still living in the depauperate creek below Lake Samsonvale, and lungfish are still to be seen in the spillway pool below Lake Wivenhoe, unable to escape now that water levels have fallen. Members of the public and local conservation groups, who used to rescue stranded fish in the past, are excluded from access to spillway pools.

Attempts to restore the riverine environment of the lungfish are now being tried. However, the situation in the few remaining habitats of the living lungfish is not encouraging and unlikely to be solved by transplanting eel grass to some denuded parts in small areas of the river. Lungfish do not eat eel grass, and it is only one of several water plants that lungfish use for spawning. What is needed is to restore the snails and clams that lungfish use for food throughout the habitat, animals that provide the major source of nutrition for the fish.

Concerned individuals have suggested that fish raised in commercial hatcheries can be released into rivers and reservoirs to supplement the natural, ageing, population [46].

However, if hatchery fish were to be released into rivers and reservoirs, they would face the same problems of environmental degradation and similar difficulties arising from lack of recruitment as the wild lungfish, that is, if they ever learned how to live in a wild environment without food supplies on tap and guaranteed shelter. Fish produced by hatcheries are unlikely to survive or to spawn if released into a wild environment. There is no suitable food any more, no refuges for young fish, and no spawning sites. Hatchery-reared fish cannot be used to supplement existing wild populations. Even if the young fish were healthy and had been well fed before release, the problem lies with the environment. Until the environment is restored throughout the catchment of the rivers, recruitment will continue to fail.

Altered habitats are now a major part of the coastal river systems. With no recruitment in reservoirs in the Brisbane and Pine Rivers, and apparently in the remaining unaltered reaches of the rivers, lungfish will soon die out in these catchments. Extinction may also be happening in the Mary and Burnett Rivers, equally affected by the building of reservoirs and by droughts and floods. Lungfish have already become extinct in Enoggera Reservoir, although a few adult fish may be surviving in the unsuitable habitat of Enoggera Creek below the reservoir wall, where spawning is unlikely to be successful. The only suggestion offered by the authorities to save the lungfish of the Brisbane River catchment and Enoggera Reservoir was to try to restore the spawning grounds in the Brisbane River below Lake Wivenhoe, which were abandoned by the lungfish over 10 years ago during the drought. However, restoration of spawning grounds in the river, even if they are successful, will only last until the next flood or dry spell. SEQwater officials have insisted that it will not be possible to improve the environment in the water impoundments because water levels cannot be kept steady. Government authorities have been made aware that recruitment has ceased in their water impoundments, and why it has happened. They know that the solution is to clean up and improve the environments in the reservoirs and to restore the spawning areas. They insist that this is not possible because the level of water in the reservoirs cannot be held at a steady level.

The story of wild Australian lungfish, *Neoceratodus forsteri*, in the natural environments of the small part of southeast Queensland where they have survived is not a happy one. There is no uplifting ending to the tale and little chance that the lungfish, the sole living member of an extensive fossil dipnoan fauna in Australia, will survive in the wild for much longer. This is not the result of the natural environmental pressures that resulted in the extinction of most of the fossil lungfish relatives throughout geological history, from the early Devonian until the more recent losses of the lungfish species of the Pleistocene. The potential demise of the Australian lungfish has a more prosaic cause. If we lose the last living representative of its group, the Neoceratodontidae, it will be because of human-induced degradation of their fragile environment, made worse by frequent flooding and lengthy droughts, as well as by Government intransigence and obstinacy. The result? No food for adults or young, no decent spawning sites, no refuges for vulnerable eggs and hatchlings, and no recruitment. Lungfish are fighting the long defeat.

**Funding:** The research did not receive any financial assistance from funding agencies in the public, commercial, or not-for-profit sectors.

**Institutional Review Board Statement:** Living lungfish described in this paper were collected with permission from the University of Queensland Animal Ethics Committee, approval number CMM/013/03/ARC, and the Queensland Fisheries Management Authority, permit number PRM03012K.

**Data Availability Statement:** Not applicable.

**Acknowledgments:** My thanks to anonymous, who invited me to contribute to the special issue on "Evolution and Diversity of Fishes in Deep Time", and to JRR Tolkien, for his concept of the long defeat to describe fighting against ultimately insuperable odds. Some of the illustrations have been used in other publications by the author, in other contexts.

**Conflicts of Interest:** The author declares no conflict of interest.

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
