# Peer review of "Lungfish and the Long Defeat"

_diversity, doi:10.3390/d15010063_

Round 1
Reviewer 1 Report
I enjoyed reading this article and am looking forward to seeing it in print. I absolutely appreciated the section discussing tooth plate development and in vivo wear. This had direct implications on other parts of the manuscript, as the tooth plate information was used to support observations of the fossil record. Just about all of my editorial comments were related to grammar and punctuation, which I think will improve the readability of the paper. I hope that my chicken scratch handwriting is legible.

Author Response
I have added comments in the text. They are highlighted in bold in the manuscript.

Reviewer 2 Report
Please seem comments on the edited pdf

Author Response

(The authors gave the same response as above.)

Reviewer 3 Report
The paleoenvironmental interpretation appears sometimes not so sound.
No taphonomic details for the fossils…and when you are dealing with erosion/attrition the author must discuss biostratinomical processes.
Diet interpretation vs tooth-plate morphology is not convincing. Crushing vs grinding…I am not very familiar with this fishes but in actinopterygians or you are a crusher or you are a grinder because there is a totally different jaws mechanics.
Maybe too long and redundant.
see also attached commented pdf

Author Response

(The authors gave the same response as above.)
